# Identifying Early Diagnostic Biomarkers Associated with Neonatal Hypoxic-Ischemic Encephalopathy

**DOI:** 10.3390/diagnostics11050897

**Published:** 2021-05-18

**Authors:** Inn-Chi Lee, Swee-Hee Wong, Xing-An Wang, Chin-Sheng Yu

**Affiliations:** 1Division of Pediatric Neurology, Department of Pediatrics, Chung Shan Medical University Hospital, Taichung 40201, Taiwan; a7710355@yahoo.com.tw; 2Institute of Medicine, School of Medicine, Chung Shan Medical University, Taichung 40201, Taiwan; i528528528@yahoo.com.tw; 3Division of Neonatology, Department of Pediatrics, Chung Shan Medical University Hospital, Taichung 40201, Taiwan; 4Department of Information Engineering and Computer Science, and Master’s Program in Biomedical Informatics and Biomedical Engineering, Feng Chia University, Taichung 40201, Taiwan; yucs@fcu.edu.tw

**Keywords:** newborn, hypoxic-ischemic encephalopathy, biomarker, MRI, lactate

## Abstract

Background: Identifying an effective method for the early diagnosis of neonatal hypoxic-ischemic encephalopathy (HIE) would be beneficial for effective therapies. Methods: We studied blood biomarkers before 6 h after birth to correlate the degree of neonatal HIE. A total of 80 patients were divided into group 1 (mild HIE) and group 2 (moderate or severe HIE). Then, 42 patients from group 2 received hypothermia therapy and were further divided into group 3 (unremarkable or mild MRI results) and group 4 (severe MRI results). Results: Between groups 1 and 2, lactate, creatinine, white blood cells, and lactate dehydrogenase (LDH) were significantly different. Between groups 3 and 4, lactate, prothrombin time, and albumin were significantly different. Sarnat staging was based on our observation that more than 45 mg/dL of lactate combined with more than 1000 U/L of LDH yielded the highest positive predictive value (PPV) (95.7%; odds ratio, 22.00), but a low negative predictive value (NPV) for moderate or severe HIE. Using more than 45 mg/dL of lactate yielded the highest NPV (71.4%) correlated with moderate or severe HIE. Conclusions: Lactate combined with LDH before 6 h after birth yielded a high PPV. Using combined biomarkers to exclude mild HIE, include moderate or severe HIE, and initialize hypothermia therapy is feasible.

## 1. Introduction

Birth asphyxia is a physiological condition observed in newborns as a result of a prolonged or profound discrepancy between oxygen demand and delivery [1]. Birth asphyxia ranges from mild to severe; however, moderate or severe birth asphyxia can cause irreversible cerebral cell damage and death, leading to hypoxic-ischemic encephalopathy (HIE). HIE may lead to an altered conscious state, autonomic instability, absence of primitive reflexes, seizures, and death. Worldwide, birth asphyxia accounts for 24% of neonatal deaths and approximately 800,000 deaths annually in individuals under the age of 5. An estimated 4 million of the 130 million newborns worldwide experience asphyxia every year, an estimated 1 million die, and an estimated 1 million develop severe, long-term sequelae [2,3,4]. HIE is a major common etiology for neonatal death and neurodevelopmental consequences.

Hypothermia as a rescue therapy for neonatal HIE is proven to be effective and to result in few adverse effects in newborns [5,6,7]. Although therapeutic hypothermia is employed to reduce neurological injury caused by HIE, a 45–55% risk of death or moderate or severe disability in treated infants remains [5,6,8]. Therapeutic hypothermia may be established as a standard treatment for HIE. Early hypothermia rescue therapy has increased the pressure on clinicians to make accurate assessments of neonatal HIE and its severity [9].

### 1.1. Side Effects of Hypothermia Rescue Therapy

Despite hypothermia rescue therapy being proven to attenuate moderate or severe neonatal HIE in newborns with mild HIE, the benefits have not been proven. Hypothermia rescue therapy may not be suitable for newborns with mild HIE because of possible coagulation disorders. In addition, therapeutic hypothermia is associated with an increased risk of intracranial hemorrhage and cardiopulmonary instability [5,10,11]. Fortunately, hypothermia therapy-associated adverse effects are mostly minor and are not associated with low temperatures [5]. Although severe side effects are uncommon, the bleeding tendency during the aggravation by a traumatic delivery, with subgaleal hematoma being a prominent example, should be examined. Subgaleal hemorrhage (SGH) is associated with severe instability of hemodynamic function, disorder of coagulation, and even mortality. The outcomes in neonatal HIE associated with SGH have not been well explored [11]. Currently, therapeutic hypothermia for stage I HIE has not resulted in significant long-term benefits [12,13]. In one study [14], 20% of newborns with perinatal acidemia, associated with only mild HIE, had abnormal short-term outcomes. Developing adjunctive assessment tools or identifying biomarkers of fetal acidemia are essential for recommending therapeutic hypothermia.

### 1.2. Biomarkers Associated with Outcomes

Common biomarkers from blood include lactate, lactate dehydrogenase (LDH), troponin-T, creatine phosphokinase (CK), the urine lactate/creatinine (L/C) ratio, and serum interleukin (IL)-10 [9,15]. Elevated serum IL-10 is associated with severe neonatal encephalopathy and adverse early childhood outcomes but is not widely observed. The urine L/C ratio might be useful for differentiating newborns with moderate or severe HIE from those with mild HIE; however, collecting urine is not immediately feasible after birth because of potential renal damage. Furthermore, serum IL-6 levels and the L/C ratio might be useful for predicting disability and mortality in newborns with HIE [16]. Acylcarnitine profiles in neonatal HIE have demonstrated the value of butyrylcarnitine as a prognostic marker [17]. Absolute lactate values can be a used as an auxiliary marker in the early estimation of long-term outcomes in newborns with neonatal asphyxia undergoing therapeutic hypothermia [18]. Troponin-T concentration exhibited favorable predictive accuracy for mortality before discharge [19]. In addition, troponin I elevation is a biomarker of myocardial ischemia in adults and children. Myocardial ischemia may be related to multiorgan injury in patients with neonatal HIE [20]. The methods used to determine the potential need for hypothermia rescue therapy include electroencephalography (EEG), amplitude-integrated EEG (aEEG), and brain sonography. In those patients with neonatal HIE, lactate levels have a negative effect on serum level of total bilirubin that may work as a natural antioxidant as a protecting effect against oxidative stress [21]. Studies have revealed other biomarkers such as neuron-specific enolase, S100B, ubiquitin carboxy-terminal hydrolase-L1, total Tau and cytokines (e.g., IL-1β, IL-6, IL-10, IL-13), tumor necrosis factor alpha, and brain-derived neurotrophic factors are helpful for diagnosing HIE and predicting neurodevelopmental outcomes [22,23,24,25] Unfortunately, tests for most biomarkers are not widely available, and studies evaluating candidate brain injury biomarkers have revealed that no single biomarker is currently in use.

### 1.3. Magnetic Resonance Imaging and Outcomes

The infants in the clinical encephalopathic group presenting with HIE required therapeutic hypothermia before 6 h after birth. Magnetic resonance imaging (MRI) of the infants’ brains revealed parenchymal abnormalities, which are consistent with hypoxic-ischemic cerebral injury. The abnormalities evident in early MRI images are highly correlated with late neurodevelopmental outcomes, particularly of the thalamus and basal ganglion. A diffuse brain MRI revealing cyst encephalomalacia and white matter injury after hypothermia rescue therapy is an indicator of poor neurodevelopmental outcomes. Other abnormalities such as deep medullary vein thrombosis, venous infarctions, and punctate hemorrhages are correlated with more favorable outcomes [11].

Early diagnosis and rapid treatment are critical for long-term prognosis in patients with neonatal encephalopathies. Identifying an effective method for the early diagnosis of neonatal HIE would be beneficial for developing effective therapies. We investigated an effective and feasible approach and combined multiple biomarkers and clinical signs to predict and improve outcomes of early therapeutic hypothermia.

## 2. Materials and Methods

### 2.1. Patients (Newborns at High Risk of Encephalopathies)

We retrospectively identified patients with neonatal HIE on the basis of a history of fetal distress, metabolic acidosis, or dependent positive-pressure ventilation immediately after birth from 2015 to 2020 at Chung Shan Medical University Hospital, a medical center in central Taiwan. HIE was classified according to Sarnat staging, with stages I, II, and III indicating mild, moderate, and severe HIE, respectively [5,6,26]. Based on HIE classification, additional examinations were performed, including complete blood count and biochemical analyses involving blood gas, liver function (glutamic oxaloacetic transaminase (GOT) and glutamic-pyruvic transaminase (GPT)), renal function (blood urea nitrogen (BUN) and creatinine), coagulation (prothrombin time (PT) and activated partial thromboplastin time (aPTT)), lactate, LDH, sodium (NA), potassium (K), glucose, and albumin. For patients in Sarnat stages II and III, a series of examinations, including head ultrasound (HUS), MRI, aEEG, continuous neonatal conventional EEG monitoring, and hearing tests (auditory brainstem response (ABR) and automated ABR), were performed after hypothermia therapy and before discharge. For patients in stage I, a series of HUSs were performed immediately at birth and at 1, 3, 7, and 14 days old. To determine differences in blood biomarkers, we divided patients in group 1 classified as stage I (mild HIE) and in group 2 classified as stages II and III (moderate and severe HIE) according to the recommendations of an experienced pediatric neurologist and a consulting neonatologist, and then compared the differences in blood biomarkers between the two groups (Figure 1).

### 2.2. Severe and Mild MRI Changes Correlated with Outcomes

To identify biomarkers correlated with MRI changes, we divided patients receiving hypothermia rescue therapy into two groups according to the severity of MRI changes. Group 3 patients presented with mild MRI changes (e.g., mild intraventricular hemorrhage, subdural hemorrhage, miniscule localized infarction, or hemorrhage of brain parenchyma), and group 4 patients presented with severe MRI changes (e.g., thalamus or basal ganglion lesions, multicystic encephalomalacia, or diffuse white matter injury).

All patient charts were retrospectively reviewed. Ethical approval was obtained from the Institutional Review Board of Chung Shan Medical University Hospital (IRB #: CS13036).

### 2.3. Evaluation of Neurodevelopmental Outcomes

Neurodevelopmental outcomes were evaluated at the corrected age of 1 year through clinical evaluations or by using the Bayley Scales of Infant and Toddler Development, Third Edition (Bayley-III).

### 2.4. Statistical Analysis

Statistical differences between groups were identified using an independent *t* test in SPSS (version 14.0; SPSS Institute, Chicago, IL, USA). Significant differences were evaluated using an independent *t* test or a chi-square test. Significance was set at *p* < 0.05. If sample distribution was nonparametric, a Mann–Whitney *U* test was performed.

## 3. Results

### 3.1. Demographic Data

Overall, 80 patients had HIE, with 25 in group 1 and 42 in group 2. In total, 13 patients were excluded due to congenital anomalies (*n* = 6), premature birth at fewer than 36 weeks (*n* = 6), or confirmed genetic defects (*n* = 1) (Figure 1). Among the 25 patients in group 1, 1 patient had EEG seizures (nonclinical) and was subsequently diagnosed as having diffuse sinus thrombosis in the superficial great sinus vein (according to MRI). Table 1 presents the demographic data of the two groups. The birth weight, sex, age, transfer mode (inborn or outborn) and method of delivery were not significantly different between the groups (Table 1).

### 3.2. Differences in Blood Biomarkers between Groups 1 and 2

In terms of the blood biomarkers of groups 1 and 2, differences were observed in lactate (group 1: 47.0 ± 25.2 mg/dL, group 2: 91.4 ± 52.9 mg/dL; *p* = 0.001), creatinine (group 1: 0.8 ± 0.2, group 2: 1.0 ± 0.2; *p* = 0.005), white blood cells (WBCs) count (group 1: 16,941.0 ± 4967.0, group 2: 23,096.9 ± 10,585.4; *p* = 0.014), and LDH (group 1: 681.5 ± 399.6, group 2: 1557.8 ± 1787.8; *p* = 0.040) (Table 2). Other biomarkers such as platelets, age, hemoglobin, SGOT, SGPT, PT, aPTT, BUN, albumin, CK, and CK myocardial band (CK-MB), were not significantly different between groups 1 and 2.

### 3.3. MRI Findings for Patients in Stages II and III after Hypothermia Rescue Therapy

Of the 28 stage II patients with HIE, 12 (42.9%) had unremarkable MRI results. Of the 14 stage III patients, 12 (85.7%) had thalamus or basal ganglion lesions, 2 (14.3%) had brain stem lesions, 1 (7.1%) had small parenchymal lesions, 3 (21.4%) had diffuse white matter injury (leukomalacia), and 3 (21.4%) had multicystic encephalomalacia. In addition, 1 (7.1%) had subdural hemorrhage and 1 (7.1%) had small parenchymal lesion (infarction or hemorrhage). Of the 28 stage II patients, 4 (14.3%) had thalamus or basal ganglion lesions, 1 (3.6%) had multicystic encephalomalacia, 3 (10.7%) had diffuse white matter injury, 6 (21.4%) had small parenchymal lesion (infarction or hemorrhage), and 7 (25.0%) had subdural hemorrhage. Compared with stage II patients, stage III patients had a significant difference (*p* < 0.05) in thalamus or basal ganglion lesions and multicystic encephalomalacia (Table 3).

### 3.4. Differences in Biomarkers Correlated with MRI Results after Hypothermia Rescue Therapy

Lactate levels in groups 3 and 4 were significantly different (group 3: 72.0 ± 40.4, group 4: 112.7 ± 57.6; *p* = 0.011), as well as PT (group 3: 15.4 ± 4.0, group 4: 21.4 ± 11.7; *p* = 0.031) and albumin (group 3: 3.7 ± 0.4, group 4: 3.3 ± 0.5; *p* = 0.035). Other biomarkers such as LDH, creatinine, platelets, hemoglobin, SGOT, SGPT, aPTT, BUN, CK, and CK-MB were not significantly different between groups 3 and 4 (Table 4).

### 3.5. Identifying Clinically useful Biomarkers

To determine useful factors for early identifying groups 1 and 2, single factors and combined factors were employed to predict sensitivity, specificity, PPV, and NPV. We chose suitable values for lactate, LDH, WBCs count, and creatinine to obtain a high PPV and NPV. We also used single factors such as lactate (>45 mg/dL), creatinine (>0.9 mg/dL), LDH (>1000 U/L), and WBCs (>20,000 mm^3^ µL) and combined factors such as lactate (>45 mg/dL) + creatinine (>0.9 mg/dL) (*n* = 26), lactate (>45 mg/dL) + WBCs count (>20,000 mm^3^ µL) (*n* = 25), and lactate (>45 mg/dL) + LDH (>1000 U/L) (*n* = 23) to predict staging. Lactate (>45 mg/dL) + LDH (>1000 U/L) (*p* = 0.000; odds ratio, 22.00) had the highest PPV (95.7%), but a low NPV and sensitivity (50.0%). Lactate (>45 mg/dL) had the highest NPV (71.4%) (Table 5 and Figure 2).

## 4. Discussion

A significant contribution of this study is its delineation of combined biomarkers to exclude patients with mild HIE and include patients with moderate or severe HIE when determining whether to initialize early hypothermia rescue therapy. This is beneficial for clinicians who want to initiate hypothermia rescue therapy and neuroprotection drug treatments earlier than is typical. Lactate, creatinine, LDH, and WBCs count may be helpful for identifying an appropriate time to initialize early hypothermia rescue therapy, but they exhibited low PPV and specificity. We demonstrated the combined biomarkers to identify stage II and III patients with HIE. For the groups who received hypothermia rescue therapy, lactate, PT, and albumin were correlated with severe MRI results, particularly in the thalamus and basal ganglion, which are strongly correlated with poor neurodevelopmental outcomes. To avoid such outcomes, aggressive treatment of hypothermia on hemostasis and low albumin should be considered.

Reports of biomarkers being correlated with MRI-indicated lesions are limited. Lactate may be beneficial for predicting severe MRI-indicated lesions that are typically correlated with poor neurodevelopmental outcomes at 1 year old [15,27]. Neurodevelopmental outcomes at 1 year old are closely related to the two weeks of MRI findings after birth for neonatal patients with stage II and stage III HIE [28,29]. Based on our observations, abnormal MRIs of thalamus and basal ganglion were strongly correlated with abnormal neurodevelopmental outcomes at 1 year old in our patients. Lactate levels in the first 6 h after birth can indicate the severity of neonatal HIE. The PT related to MRI-indicated lesions indicates the severity of HIE caused by coagulation defects. However, we did not observe that aPTT significantly contributed to MRI-indicated lesions. Albumin level plays a role in MRI-indicated lesions, suggesting that a low albumin level indicates neonatal hypotension, which may cause secondary damage. Albumin should be transfused to elevate blood pressure, which is attributed to HIE and hypothermia.

SGH has been classified as a relative contraindication, but not an absolute contraindication for hypothermia [11,25]. In addition, SGH is potentially lethal to newborns, causing shock, coagulopathy, disseminated intravascular coagulation, anemia, renal or hepatic injury, and lactic acidosis [11]. SGH concurrent with HIE is difficult to manage and may cause morbidity and mortality [11]. Changes in head circumference between the first and second measurements within 48 h after birth is not an indicator of SGH and HIE [11]. SGH is often either non-hemorrhagic or results in small amounts of blood products. The suggested underlying mechanisms of SGH include suture diastasis, skull fracture, and ruptured emissary vein secondary to fragmentation of the superior margin of the parietal bone [11]). In the current study, we determined that PT is a risk factor for severe MRI changes. Due to coagulation, aggressive transfusion is suggested for correcting blood coagulation.

In the current study, the strongest biomarkers were, in order of strength, lactate, creatinine, WBCs count, and LDH. In terms of efficacy, using a single biomarker is typically unreliable. Therefore, using combined biomarkers in addition to clinical signs is suggested. A useful clinical sign is crying and intact facial expression in those patients with HIE; however, further study is required. Determining the degree of neonatal HIE according to a single biomarker is not feasible because the brain structure is highly complex in terms of functional connectivity. Early crying after birth could be a critical sign for excluding severe HIE. A delay in crying may be associated with conditions such as HIE, anesthesia, and neuromuscular disease. A 2-h delay in crying requires immediate evaluation and treatment. Early crying and intact facial expression could be an early and reliable sign against neonatal HIE, and these manifestations are easily confirmed by clinicians. Failure to cry in newborns should be considered a critical sign of HIE because it indicates impairments in the pathway from the brain stem to the thalamus and cortex. Accordingly, combining clinical signs and biomarkers is a suggested method for clinicians judging newborns. We proposed a feasible approach for treating neonatal HIE (Figure 3).

This study had several limitations. A limited number of patients with HIE were analyzed to identify risk factors; hence, our findings might be biased. The data also had some HIE-related bias because of the low number of patients. Therefore, studies analyzing more patients are warranted. Furthermore, in terms of the stage I group of patients with HIE with favorable outcomes, a comprehensive imaging study was not covered by National Health Insurance in Taiwan. However, a series of HUSs can support imaging findings, and a clinical follow-up of up to 1 year could confirm that the patients did not have major significant brain lesions.

## 5. Conclusions

Blood lactate levels, creatinine, WBCs count, and LDH are reliable biomarkers because they were observed to be significantly different in newborns with mild HIE and in newborns with moderate or severe HIE. Considering lactate levels and LDH together yielded a high PPV and specificity. Because lactate exhibited a high NPV, it can be used to exclude patients with mild neonatal HIE. This combined method can be used to exclude patients with mild HIE and include patients with moderate or severe neonatal HIE, helping to identify an appropriate time to initialize early hypothermia rescue therapy.

## Figures and Tables

**Figure 1 diagnostics-11-00897-f001:**
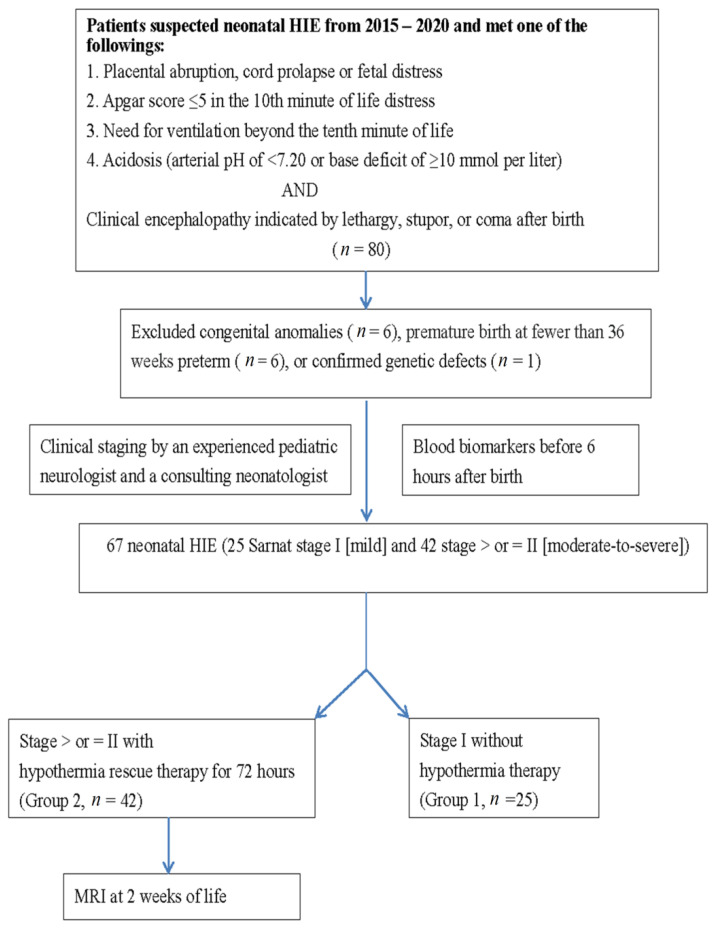
Flow chart of the study procedure. HIE indicates hypoxic-ischemic encephalopathy; MRI, magnetic resonance imaging.

**Figure 2 diagnostics-11-00897-f002:**
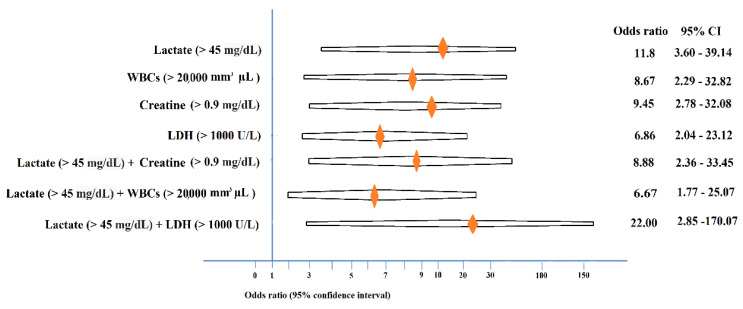
Odds ratio of different biomarkers for determining Sernat staging of patients with neonatal HIE.

**Figure 3 diagnostics-11-00897-f003:**
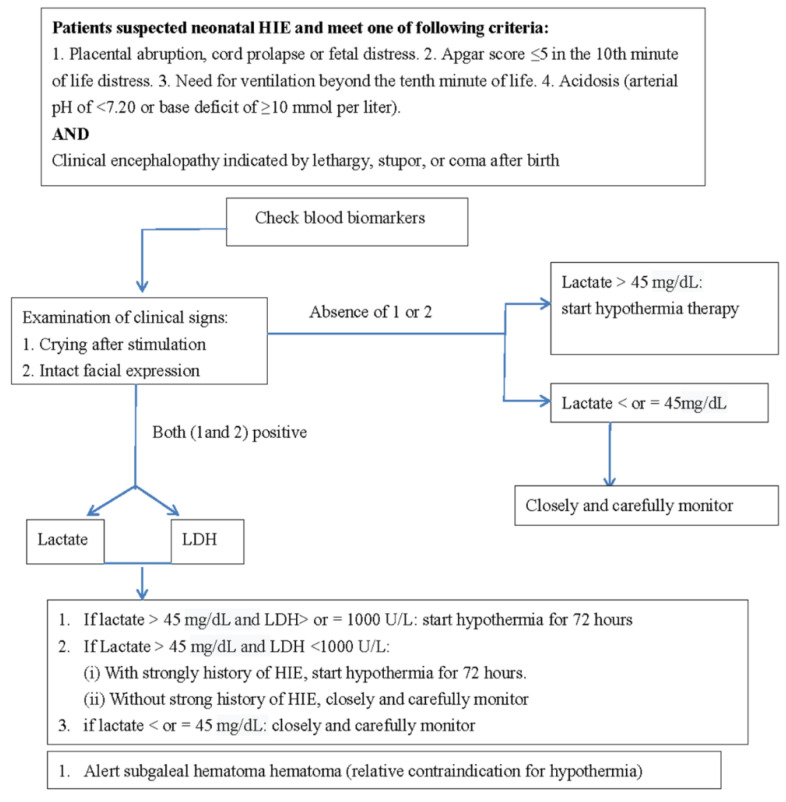
Feasible approach for treating patients with neonatal HIE.

**Table 1 diagnostics-11-00897-t001:** Demographic data of 67 newborns with HIE.

Variables	Hypoxic-Ischemic Encephalopathy, Stage I; (*n* = 25)	Hypoxic-Ischemic Encephalopathy Stage II and III, (*n* = 42)	*p ^#^*
Gestational age (weeks)	39.3 ± 1.7	39.7 ± 1.3	NS
Birth weight (gm)	3421 ± 520	3530 ± 576	NS
Gender			NS
Male	15 (60.0%)	26 (61.9%)	
Female	10 (40.0%)	16 (38.1%)	
Transfer mode			NS
Inborn	9 (36.0%)	15 (35.7%)	
Outborn	16 (64.0%)	27 (64.3%)	
Method of delivery			NS
Cesarean section	11 (44.0%)	15 (35.8%)	
Vaginal delivery	14 (56.0%)	27 (64.2%)	

**^#^** NS indicates *p* value > 0.05 and non-significant result.

**Table 2 diagnostics-11-00897-t002:** Differences in blood biomarkers between group 1 (mild HIE) and group 2 (moderate and severe HIE).

Biomarkers	Group 1	ST	Group 2	ST	*p* Value
**WBCs *** (9100–34,000 mm^3^ µL)	**16,941.0**	**4967.0**	**23,096.9**	**10,585.4**	**0.014 ***
Platelet (84–478 mm^3^ µL)	240,523.8	86,145.0	225,285.7	73,186.8	0.466
Hemoglobin (13.88 ± 1.34 g/dL)	16.4	2.2	17.7	7.0	0.426
SGOT (30–100 U/L)	72.1	34.6	225.6	319.8	0.062
SGPT (6–40 U/L)	18.0	10.7	68.9	104.6	0.051
BUN (3–12 mg/dL)	9.6	2.9	11.4	3.9	0.077
**Creatinine *** (0.03–0.50 mg/dL)	**0.8**	**0.2**	**1.0**	**0.2**	**0.005 ***
**Lactate **** (4.4 to 14.4 mg/dL)	**47.0**	**25.2**	**91.4**	**52.9**	**0.001 ****
**LDH *** (170–580 U/L)	**681.5**	**399.6**	**1557.8**	**1787.8**	**0.040 ***
PT (13.0 ± 1.43 s)	14.3	3.6	18.2	9.0	0.071
aPTT (42.9 ± 5.80 s)	56.0	19.2	63.3	28.2	0.312
Albumin (2.5–3.4 g/dL)	3.5	0.8	3.5	0.5	0.706
Glucose (40–60 mg/dL)	97.1	30.1	131.5	81.5	0.067
Na (133–146 mmol/L)	135.9	3.6	135.6	3.6	0.769
K (3.2–5.5 mmol/L)	4.1	0.5	4.1	0.8	0.939
CK (39–308 U/L)	1355.7	981.0	3250.5	4454.9	0.090
CK-MB (0–4.5 ng/mL)	39.5	27.2	73.5	99.2	0.236

***** Bold fonts indicate *p* < 0.05; ******
*p* < 0.005. HIE, hypoxic-ischemic encephalopathy; ST, standard deviation; WBCs, white blood cells; GOT, aspartate transaminase; GPT, alanine transaminase; BUN, blood urea nitrogen; LDH, lactate dehydrogenase; PT, prothrombin time; aPTT, activated partial thromboplastin time; CK, creatine phosphokinase; CK-MB, creatine kinase Mb; K, potassium; Na, sodium.

**Table 3 diagnostics-11-00897-t003:** Brain MRI results for stage II and stage III patients in group 2 after hypothermia rescue therapy.

Brain Lesions on MRI	Stage II * (*n* = 28)	Stage III * (*n* = 14)	*p* Value ^#^
**Thalamus or basal ganglion**	4 (14.3%)	12 (85.7%)	**0.000 ****
Brain stem (one of midbrain, pons, and medulla)	0 (0.0%)	2 (14.3%)	NS
**Multicystic encephalomalacia**	1 (3.6%)	4 (28.8%)	**0.038 ***
Diffuse white matter injury	3 (10.7%)	3 (21.4%)	NS
Small parenchymal lesion (infarction/hemorrhage)	6 (21.4%)	1 (7.1%)	NS
Subdural hemorrhage	7 (25.0%)	1 (7.1%)	NS
**Unremarkable**	12 (42.9%)	0 (0.0%)	**0.003 ****

***** Bold fonts indicate *p* < 0.05; ******
*p* < 0.005; MRI indicates magnetic resonance imaging; ***** Some patients had multiple brain lesions in MRI; ^#^ NS indicates *p* value > 0.05 and non-significant result.

**Table 4 diagnostics-11-00897-t004:** Differences in blood biomarkers correlated with MRI results after hypothermia rescue therapy in 42 patients with neonatal HIE.

Biomarkers	Group 3 ^±^ (*n* = 22)	Standard Deviation	Group 4 ^±^ (*n* = 20)	Standard Deviation	*p* Value
WBCs	21,744.1	8130.7	24,584.9	12,817.2	0.392
Platelet	232,454.5	73,365.2	217,400.0	74,056.5	0.512
Hemoglobin	16.9	2.1	18.5	10.0	0.454
SGOT	164.6	197.7	292.8	410.4	0.198
SGPT	54.7	92.2	84.5	117.2	0.364
BUN	10.8	3.1	12.1	4.7	0.304
Creatinine	1.0	0.2	1.0	0.2	0.974
**Lactate ***	**72.0**	**40.4**	**112.7**	**57.6**	**0.011 ***
LDH	1439.3	1625.5	1682.2	1978.9	0.669
**PT ***	**15.4**	**4.0**	**21.4**	**11.7**	**0.031 ***
aPTT	60.9	21.2	65.8	34.8	0.580
**Albumin ***	**3.7**	**0.4**	**3.3**	**0.5**	**0.035 ***
Glucose	109.5	65.7	154.7	91.3	0.075
Na	135.9	3.7	135.3	3.6	0.538
K	4.2	0.8	4.0	0.8	0.444
CK	4179.9	5502.5	2174.4	2549.8	0.153
CK-MB	68.5	82.4	78.5	116.9	0.802

^±^ Group 3 patients presented with mild MRI changes (e.g., mild intraventricular hemorrhage, subdural hemorrhage, miniscule localized infarction, or small hemorrhage of brain parenchyma), and group 4 patients presented with severe MRI changes (e.g., thalamus or basal ganglion lesions or multicystic encephalomalacia or diffuse white matter injury); * Bold fonts indicate *p* < 0.05; MRI, magnetic resonance imaging; HIE, hypoxic-ischemic encephalopathy; WBCs, white blood cells; GOT, aspartate transaminase; GPT, alanine transaminase; BUN, blood urea nitrogen; LDH, lactate dehydrogenase; PT, prothrombin time; aPTT, activated partial thromboplastin time; CK, creatine phosphokinase; CK-MB, creatine kinase Mb; K, potassium; Na, sodium.

**Table 5 diagnostics-11-00897-t005:** Application of single and combined biomarkers to determine the odds ratio, PPV, NPV, sensitivity, and specificity.

Blood Biomarkers							95% CI	
	PPV (%)	NPV (%)	Specificity (%)	Sensitivity (%)	*p* Values	Odds Ratio	Lower	Upper
Lactate (>45 mg/dL) (*n* = 45)	82.6	71.4	65.2	86.4	0.000	11.88	3.60	39.14
WBC (>20,000 mm^3^ µL) (*n* = 29)	89.7	50.0	86.4	57.8	0.001	8.67	2.29	32.82
Creatinine (>0.9 mg/dL) (*n* = 31)	87.1	58.3	84.0	64.3	0.000	9.45	2.78	32.08
LDH (>1000 U/L) (*n* = 30)	86.7	51.4	82.6	59.1	0.001	6.86	2.04	23.12
Lactate (>45 mg/dL) + Creatinine (>0.9 mg/dL) (*n* = 26)	88.5	52.5	87.5	54.8	0.000	8.88	2.36	33.45
Lactate (>45 mg/dL) + WBCs (>20,000 mm^3^ µL) (*n* = 25)	88.0	47.6	87.0	50.0	0.003	6.67	1.77	25.07
Lactate (>45 mg/dL) + LDH (>1000 U/L) (*n* = 23)	95.7	50.0	95.7	50.0	0.000	22.00	2.85	170.07

HIE, hypoxic-ischemic encephalopathy; PPV, positive prediction rate; NPV, negative prediction rate; CI, confidence interval; WBCs, white blood cells; LDH, lactate dehydrogenase.

## Data Availability

The datasets used and/or analyzed during the current study are available from the corresponding author on request.

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
