# Peer review of "Identifying Early Diagnostic Biomarkers Associated with Neonatal Hypoxic-Ischemic Encephalopathy"

_diagnostics, 2021, doi:10.3390/diagnostics11050897_

Round 1
Reviewer 1 Report
The article is an original research, the design is well planned, the material is structured and well presented. The remarks relate to the presentation of numerical values in tables 2, 4; the number of significant digits and the rounding rules should be rechecked. I did not find a discussion of the violations identified at 1 year old and their relationship with the biochemical parameters described in the article. Only the results of the examination after 2 weeks are discussed and only for stages 2 and 3. I think this information should be added to the article.
Author Response
- Reviewer 1: Comments and Suggestions for Authors
- The article is an original research, the design is well planned, the material is structured and well presented. The remarks relate to the presentation of numerical values in tables 2, 4; the number of significant digits and the rounding rules should be rechecked.
Reply: Thanks for the valuable opinions. We have rechecked numerical values in tables 2 and 4. The number of significant digits and the rounding rules have been be rechecked.
- I did not find a discussion of the violations identified at 1 year old and their relationship with the biochemical parameters described in the article. Only the results of the examination after 2 weeks are discussed and only for stages 2 and 3. I think this information should be added to the article.
Reply: Thanks for the valuable opinion. Two weeks of MRI after birth in neonatal HIE receiving hypothermia therapy is closely related to the neurodevelopmental outcome at 1 year old [28,29]. The abnormal MRI, particularly in the thalamus and basal ganglion, is strongly correlated with poor neurodevelopmental outcomes. The comprehensive evaluation of neurodevelopmental outcomes at 1 years old will be further addressed in the future in the study. We added in Discussion, paragraph 2.
In Discussion, paragraph 2
“Reports of biomarkers being correlated with MRI-indicated lesions are limited. Lactate may be beneficial for predicting severe MRI-indicated lesions that are typically correlated with poor neurodevelopmental outcomes at 1 year old [15,27]. Lactate levels in the first 6 h after birth can indicate the severity of neonatal HIE. The PT related to MRI-indicated lesions indicates the severity of HIE caused by coagulation defects. However, we did not observe that aPTT significantly contributed to MRI-indicated lesions. Albumin level plays a role in MRI-indicated lesions, suggesting that a low albumin level indicates neonatal hypotension, which may cause secondary damage. Albumin should be transfused to elevate blood pressure, which is attributed to HIE and hypothermia.”
Changed to:
“Reports of biomarkers being correlated with MRI-indicated lesions are limited. Lactate may be beneficial for predicting severe MRI-indicated lesions that are typically correlated with poor neurodevelopmental outcomes at 1 year old [15,27]. Neurodevelopmental outcomes at 1 year old are closely to the two weeks of MRI findings after birth for neonatal patients with stage II and stage III HIE [28,29]. Based on our observations, abnormal MRIs of thalamus and basal ganglion were strongly correlated with abnormal neurodevelopmental outcomes at 1 year old in our patients. Lactate levels in the first 6 h after birth can indicate the severity of neonatal HIE. The PT related to MRI-indicated lesions indicates the severity of HIE caused by coagulation defects. However, we did not observe that aPTT significantly contributed to MRI-indicated lesions. Albumin level plays a role in MRI-indicated lesions, suggesting that a low albumin level indicates neonatal hypotension, which may cause secondary damage. Albumin should be transfused to elevate blood pressure, which is attributed to HIE and hypothermia.”
- We added two references.
28. Lin, Y.K.; Hwang-Bo, Seok.; Seo, Y.M.; Youn, Y.A. Clinical seizures and unfavorable brain MRI patterns in neonates with hypoxic ischemic encephalopathy. Medicine (Baltimore). 2021, 26, 100, e25118.
29. Chang, P.D.; Chow, D.S.; Alber, A,; Lin, Y.K.; Youn, Y.A. Predictive Values of Location and Volumetric MRI Injury Patterns for Neurodevelopmental Outcomes in Hypoxic-Ischemic Encephalopathy Neonates. Brain. Sci. 2020, 16; 10, 991.

Reviewer 2 Report
This is a simple study that compares the results of lab tests between a number of groups in patients with neonatal hypoxic-ischemic encephalopathy (HIE). The authors divided 80 HIE patients into 4 groups according to the severity. The results showed a significant difference in several measurements, such as lactate and lactate dehydrogenase (LDH), between these groups. The authors concluded that a lactate level of 45mg/dl and a LDH level of 1000 U/L are good thresholds for distinguishing the mild an severe HIE, based on the analysis of positive / negative predictive values.
This is a good study providing useful clinical data.
Author Response
- Reviewer 2: Comments and Suggestions for Authors
This is a simple study that compares the results of lab tests between a number of groups in patients with neonatal hypoxic-ischemic encephalopathy (HIE). The authors divided 80 HIE patients into 4 groups according to the severity. The results showed a significant difference in several measurements, such as lactate and lactate dehydrogenase (LDH), between these groups. The authors concluded that a lactate level of 45mg/dl and a LDH level of 1000 U/L are good thresholds for distinguishing the mild an severe HIE, based on the analysis of positive / negative predictive values. This is a good study providing useful clinical data.
Reply: Thanks for the valuable opinion.
